# Eucalyptol Ameliorates Retinal Microvascular Defects through Modulating ER Stress and Angiopoietin–Tie Signaling in Diabetic Eyes

**DOI:** 10.3390/ijms25147826

**Published:** 2024-07-17

**Authors:** Dong Yeon Kim, Sin-Hye Park, Zaee Yoon, Jimin Kim, Min-Kyung Kang, Young-Hee Kang

**Affiliations:** 1Department of Food and Nutrition, Andong National University, Andong 36729, Republic of Korea; ehddus3290@naver.com (D.Y.K.); dd0682@naver.com (Z.Y.); dksehdehdqn@naver.com (J.K.); 2Department of Food and Nutrition and Korean Institute of Nutrition, Hallym University, Chuncheon 24252, Republic of Korea; shpark88@hallym.ac.kr

**Keywords:** amyloid-β, angiopoietin, stress, eucalyptol, inner blood–retinal barrier

## Abstract

Loss of the inner blood–retinal barrier (BRB) integrity is a main feature of ocular diseases such as diabetic macular edema. However, there is a lack of clarity on how inner BRB function is modulated within the diabetic retina. The current study examined whether eucalyptol inhibited inner BRB destruction and aberrant retinal angiogenesis in 33 mM glucose-exposed human retinal microvascular endothelial (RVE) cells and db/db mice. This study further examined the molecular mechanisms underlying endothelial dysfunction including retinal endoplasmic reticulum (ER) stress and angiopoietin (Ang)/Tie axis in conjunction with vascular endothelial growth factor (VEGF). Eucalyptol is a naturally occurring monoterpenoid and an achiral aromatic component of many plants including eucalyptus leaves. Nontoxic eucalyptol reduced the production of amyloid-β (Aβ) protein in glucose-loaded RVE cells and in diabetic mice. This natural compound blocked apoptosis of Aβ-exposed RVE cells in diabetic mouse eyes by targeting ER stress via the inhibition of PERK-eIF2α-ATF4-CHOP signaling. Eucalyptol promoted activation of the Ang-1/Tie-2 pathway and dual inhibition of Ang-2/VEGF in Aβ-exposed RVE cells and in diabetic eyes. Supply of eucalyptol reversed the induction of junction proteins in glucose/Aβ-exposed RVE cells within the retina and reduced permeability. In addition, oral administration of eucalyptol reduced vascular leaks in diabetic retinal vessels. Taken together, these findings clearly show that eucalyptol inhibits glucose-induced Aβ-mediated ER stress and manipulates Ang signaling in diabetic retinal vessels, which ultimately blocks abnormal angiogenesis and loss of inner BRB integrity. Therefore, eucalyptol provides new treatment strategies for diabetes-associated RVE defects through modulating diverse therapeutic targets including ER stress, Ang-1/Tie-2 signaling, and Ang-2/VEGF.

## 1. Introduction

Retinal endothelial cells form an internal vascular lining indispensable for normal blood flow that supplies and drains the active neural retina [1]. The endothelium manages the exchange of chemical substances between blood and the retina, and maintains the inner blood–retinal barrier (BRB) by regulating vascular wall tension and angiogenesis [2]. Abnormal retinal neovascularization constitutes the most common cause of vision loss [3]. Vascular lesions observed in the retina in the course of diabetes, referred to as diabetic retinopathy (DR), are caused by vascular abnormalities in small vessels [2]. Several studies have demonstrated that excessive angiogenesis of retinal vessels results in vitreous hemorrhage, increased vascular permeability, and exudative retinal detachments, leading to vision impairment [4,5]. Notable features of retinal endothelial cells are a lack of fenestrations and the presence of specialized intercellular junctions of *zonula occludens*, leading to the formation of stable and tight unions with neighboring cells, which contributes to the BRB [1]. Also, specific endothelial adhesion molecules such as vascular endothelial (VE)-cadherin are involved in maintaining endothelial cell integrity. High glucose destroys the outer and inner BRB due to disruption of cellular junctions, causing diabetic macular edema and retinal cell degeneration [3,6,7].

Numerous studies have highlighted the potential links between Alzheimer’s disease (AD) and the abnormality of insulin signaling related to diabetes [8,9]. There is a growing consensus that hyperglycemia is a potential risk factor for the development of AD [10,11]. Diabetes and AD share several risk factors including oxidative stress, inflammation, and amyloid-β (Aβ) deposition [8]. The association between these diseases implies pathological alterations involving the deposition of vascular Aβ plaques in the neurosensory brain [12,13]. Hyperglycemia increases intercellular permeability in endothelial cells and reduces junctional proteins through the induction of amyloid precursor proteins with increased Aβ production [11]. A recent study reports that Aβ peptide induces angiogenesis in an AD mouse model through expression of placental growth factor and angiopoietin (Ang)-2 [14]. Accordingly, the interventions that directly or indirectly affect angiogenesis may have beneficial effects on amyloid peptides and other pathways related to diabetes and AD.

The disruption of normal endoplasmic reticulum (ER) function triggers a stress response, known as the unfolded protein response (UPR) [15]. ER stress is an important contributor to vascular defects as well as neurodegeneration in multiple ocular diseases [16]. ER stress causes BRB breakdown and retinal neovascularization in DR [17]. Accordingly, the advancement of therapy targeting ER stress may provide new therapeutic strategies for ocular diseases. On the other hand, a growing body of evidence has demonstrated that Aβ causes ER stress, increasing levels of several mediators of UPR signaling pathways and causing apoptotic cell death in brain endothelial cells [18,19]. Collectively, it can be assumed that glucotoxicity induces the production of Aβ, thus resulting in ER stress, which may increase the destruction of tight junctions in retinal capillary endothelial cells. A report shows that ER stress plays a role in Aβ overproduction and apoptotic pathway activation in photoreceptor cells [20]. However, the molecular mechanisms of ER stress-mediated retinal toxicity of Aβ still remain unknown. 

Several studies have reported that ER stress-targeted therapeutic strategies might be useful in DR for counteracting vascular defects of the retina [17,21,22]. Natural compounds targeting UPR components and ER stress offer a novel strategic approach to ocular diseases. Resveratrol plays a protective role in ER stress-induced retinal vascular degeneration by inhibiting ER stress [16]. Eucalyptol is a natural monoterpenoid found in many plants and essential oil fractions (Figure 1A). This compound has been studied for its pharmacological effects on oxidative stress and inflammation [23,24,25]. Our previous studies have shown that eucalyptol inhibits Aβ-mediated disruption of the outer BRB in diabetic eye and glucose-loaded retinal pigment epithelial cells [26]. The current study further investigated how eucalyptol mitigated the Aβ-mediated rupture of tight junctions on the inner BRB and aberrant retinal angiogenesis in high glucose-exposed human retinal microvascular endothelial (RVE) cells and db/db mice. This study focused on retinal ER stress by targeting the Ang/immunoglobulin-like and endothelial growth factor-like domains (Tie) axis in conjunction with vascular endothelial growth factor (VEGF).

## 2. Results

### 2.1. Inhibition of Glucose-Induced Aβ Formation by Eucalyptol

It is well documented that diabetes mellitus causes an increase in Aβ peptide levels [8]. The current study investigated how eucalyptol inhibited Aβ-induced dysfunction of RVE cells under diabetic conditions. When RVE cells were treated with 33 mM glucose or 5 μM Aβ, the cell viability was not influenced even in the presence of 20 μM eucalyptol (Figure 1C,D). To examine that prolonged high glucose enhanced endothelial formation of Aβ, human RVE cells were exposed to 33 mM glucose for 5 days. In endothelial cells stimulated by glucose for 3 days, the cellular levels of Aβ were temporally enhanced with a 60–70% increase (Figure 1E). However, such an increase was significantly diminished by supplementing 1–20 μM eucalyptol (Figure 1F). On the other hand, the FITC-immunohistochemical data revealed that Aβ level increased in the retinal tissues of diabetic mice (Figure 1G). In eucalyptol-treated mice, the elevated level of Aβ significantly declined.

### 2.2. Blockade of Apoptosis of Vascular Cells in Diabetic Retina by Eucalyptol

Hyperglycemia has been shown to promote apoptosis of RVE cells and play a key role in the pathogenesis of DR [22]. This study examined whether eucalyptol modulated the expression of the apoptosis-related proteins of bcl-2 and bax in high glucose- and Aβ-experienced RVE cells. As shown in Figure 2A, the exposure to high glucose and Aβ declined bcl-2 induction in RVE cells and enhanced bax expression. However, treatment of 1–20 μM eucalyptol restored bcl-2 expression in a dose-dependent manner. In contrast, the bax induction was significantly diminished in the eucalyptol-treated Aβ-exposed RVE cells (Figure 2A). This study also investigated whether high glucose and Aβ activated caspase 12 in RVE cells. When glucose and Aβ were treated to RVE cells for 3 days, the cleaved caspase 12 was significantly enhanced. In contrast, eucalyptol dose-dependently reduced this increase (Figure 2A).

To assess RVE cell apoptosis under hyperglycemia and the presence of Aβ, a TUNEL assay was conducted. The TUNEL staining revealed that 33 mM glucose and 5 μM Aβ highly generated many DNA fragments in RVE cells, showing that high glucose and Aβ promoted RVE cell apoptosis. However, DNA fragments were dose-dependently reduced in eucalyptol-treated cells (Figure 2B).

Consistent with the culture results of RVE cells, the in vivo animal data supported that eucalyptol blocked vascular endothelial cell apoptosis within the retina. The FITC-immunohistochemical data revealed that bcl-2 induction declined in diabetic retinal tissues of mice (Figure 3). Oral administration of 10 mg/kg eucalyptol restored the bcl-2 induction. On the contrary, the Cy3-immunohistochemical staining showed that the tissue level of bax was enhanced in diabetic eye tissue, whereas eucalyptol treatment counteracted (Figure 3).

### 2.3. Inhibition of ER Stress by Eucalyptol

Many recent studies have shown that ER stress can be related to the death of the vascular cells in the retina [27,28]. This study attempted to examine whether eucalyptol attenuated ER stress-associated apoptosis in Aβ-exposed RVE cells. High glucose and Aβ elevated the ER stress in RVE cells through increased phosphorylation of protein kinase R-like ER kinase (PERK), the ER stress transducer (Figure 4A). In addition, the activation of eukaryotic initiation factor 2 alpha (eIF2α) was accompanied in RVE cells exposed to Aβ. Furthermore, activating transcription factor-4 (ATF4) and C/EBPα-homologous protein (CHOP) involved in the downstream PERK-mediated pathway were highly activated by glucose and Aβ (Figure 4B,C). When RVE cells were supplied with 1–20 μM eucalyptol, the activation and induction of PERK and its downstream proteins of elF2α, ATF4, and CHOP were attenuated in a dose-dependent manner (Figure 4).

This study further examined the activation and induction of ER stress proteins in diabetic eyes. The in vivo animal data supported that eucalyptol blocked the occurrence of ER stress in diabetic eyes. Oral administration of 10 mg/kg eucalyptol lowered the levels of PERK, eIF2α, ATF4, and CHOP and was greatly elevated in the eye tissues of diabetic mice (Figure 5). Thus, it is deemed that eucalyptol inhibited the ER stress by blocking the PERK-mediated signaling pathway in the diabetic retina, where Aβ may be involved.

### 2.4. Blockade of Retinal Angiogenesis by Eucalyptol

Secondary to abnormal retinal blood vessel growth, increased retinal blood flow is of pathogenic importance in the progression of DR [29]. The induction of vascular endothelial growth factor (VEGF), the primary factor implicated in the alteration of retinal vascular function leading to angiogenesis, was enhanced in Aβ-loaded RVE cells (Figure 6A). In contrast, the Aβ-induced expression of VEGF was dose-dependently attenuated by treating eucalyptol.

Recent studies have proposed that Ang proteins play a role in the pathophysiology of diabetic macular edema and proliferative DR [30,31]. The current study examined whether eucalyptol attenuated Aβ-triggered Ang/tyrosine kinase with Tie pathway in diabetic retinal diseases. When RVE cells were exposed to 5 μM Aβ for 3 days, the induction of Ang-1 and Ang-2 was reciprocally influenced (Figure 6B). The reduced induction of Ang-1 was significantly enhanced in cells treated with eucalyptol, while this compound diminished the expression of Ang-2 elevated in Aβ-exposed cells (Figure 6B). Furthermore, Aβ inhibited the expression of the Tie2 receptor in RVE cells, which was restored by the supply of eucalyptol (Figure 6C).

The in vivo study attempted to confirm that eucalyptol inhibited aberrant angiogenesis and vascular destabilization in diabetic eyes. Consistent with cell culture results, increased VEGF level in diabetic eyes was demoted by treating eucalyptol (Figure 7A). Oral treatment of 10 mg/kg eucalyptol promoted the Ang-1/Tie2 pathway in diabetic eyes (Figure 7B,C). On the contrary, the retinal tissue level of Ang2 declined in eucalyptol-treated diabetic mice. Accordingly, eucalyptol may inhibit angiogenesis and improve vascular stability by applying the Ang-1/Tie-2 pathway or targeting Ang-2/VEGF in the diabetic retinal vasculature.

### 2.5. Suppressive Effect of Eucalyptol on Retinal Vascular Leakage

RVE cells lining blood vessels regulate vascular barrier function by tightly maintaining vascular permeability [32,33]. The induction of the adherence junction protein of VE-cadherin and the tight junction protein of occludin-1 was attenuated in RVE cells exposed to 5 μM Aβ, which is inhibited by eucalyptol (Figure 8A). Furthermore, oral supplementation of eucalyptol accelerated the decreased induction of VE-cadherin in diabetic eyes (Figure 8B).

To determine whether high glucose and Aβ promoted vascular permeability, the permeation of FITC-labeled BSA was examined in 33 mM glucose- or Aβ-loaded RVE cells on transwell inserts. As a result, 33 mM glucose and 5 μM Aβ increased the permeation of fluorescein isothiocyanate (FITC)-labeled BSA (Figure 8C). However, the permeability declined in the 20 μM eucalyptol-treated RVE cells.

This study further attempted to confirm that eucalyptol may block leakage by maintaining the retinal vascular permeability under diabetic conditions. This study examined the inhibitory effects of eucalyptol on retinal vascular leakage using FITC-conjugated dextran in a flat-mounted mouse retina. No vascular abnormality was observed in the retinas of db/m mice (Figure 9). In contrast, there were multiple vascular leaks in diabetic retinal vessels, as indicated in red arrows. However, diffuse staining was reduced in 10 mg/kg eucalyptol-treated retinal vasculature (Figure 9).

## 3. Discussion

Seven major findings were extracted from this study. (1) Nontoxic eucalyptol reduced Aβ protein production in glucose-loaded RVE cells and in diabetic mouse eyes. (2) Eucalyptol mitigated apoptosis of Aβ-exposed RVE cells and diabetic retinal cells. (3) Eucalyptol attenuated Aβ-mediated ER stress in RVE cells and diabetic mouse eyes by blocking the PERK-eIF2α-ATF4-CHOP signaling pathway. (4) This natural compound led to the activation of the Ang-1/Tie-2 pathway and conversely dual inhibition of Ang-2/VEGF in Aβ-exposed RVE cells and in diabetic eyes. (5) Induction of the junction proteins of VE-cadherin and occludin-1 was reduced in RVE cells exposed to Aβ, which was reversed by the supply of eucalyptol. (6) Increased permeation by Aβ was diminished in eucalyptol-treated RVE cells. (7) Oral administration of eucalyptol attenuated vascular leakages clearly observed in diabetic retinal vessels. Taken together, these findings evidently show that eucalyptol inhibits Aβ-mediated ER stress and stimulates Ang-1/Tie-2 signaling in diabetic retinal vessels, which ultimately blocks abnormal angiogenesis and loss of inner BRB integrity.

Several studies have described that the Aβ peptides increase in the eye under diabetic conditions, resulting in functional defects including apoptosis, destruction of cell junctions, and retinal degeneration in various cell types of eyes [11,20,34,35]. Accordingly, one can assume therapeutic targets between Aβ and ocular diseases. Naturally occurring compounds can be promising dietary or pharmacological candidates to handle retinal dysfunction. Our previous study has shown an increase in Aβ levels in glucose-exposed retinal pigment epithelial (RPE) cells and db/db mouse eyes, in which eucalyptol attenuates the glucose-mediated Aβ production [26]. Aβ enhanced the generation of reactive oxygen species (ROS) and induction of receptors for advanced glycation end products (RAGE) in glucose-exposed RPE cells and diabetic eyes, which was counteracted by eucalyptol. Similarly, a flavonoid puerarin attenuates Aβ-induced activation of NLRP3 inflammasome in RPE cells via inhibition of ROS-dependent oxidative and ER stresses [36]. The current study further revealed that eucalyptol inhibited Aβ-triggered apoptosis of glucose-exposed vascular endothelial cells present in the inner BRB regions.

Increasing evidence indicates that ER stress and UPR signaling are crucially involved in the pathology of ocular diseases [17,27,28]. Natural therapies targeting ER stress have been suggested to provide new treatment strategies for ocular diseases such as DR and retinitis pigmentosa. A previous study reported that chrysin improved the retinoid visual cycle by alleviating ER stress via AGE-RAGE activation in diabetic models [37]. Resveratrol alleviates ER stress-induced retinal vascular collapse through the inhibition of ER stress [16]. In this study, eucalyptol attenuated ER stress via blocking caspase-12 activation and PERK-eIF2α-ATF4/CHOP signaling in Aβ/glucose-exposed RVE cells and diabetic mice. On the other hand, other report shows that ER stress increases the production of Aβ peptides in retinal ganglion cells [20]. Accordingly, it is possible that eucalyptol may first impede ER stress in the diabetic retina, then reducing Aβ production. Unfortunately, this study did not examine the effects of tunicamycin, an ER stressor, on Aβ production in RVE cells. Since proteostasis imbalance disrupts ER function, aberrant expression of proteins forming BRB and responsible for angiogenesis can be a promising treatment target in ocular diseases [17,27].

It is well established that ER stress is associated with the loss of RVE cells, pericytes, and Müller cells that contribute to diabetic inner BRB damage [17]. In this study, Aβ promoted CHOP activation and VEGF elevation in diabetic RVE cells. In contrast, eucalyptol protected diabetic RVE cells from Aβ-induced apoptosis and VEGF induction. A polymethoxylated flavone nobiletin facilitates the repair of diabetic inner BRB disruption through the regulation of VEGF [38]. Under pathological conditions, Ang-2 is upregulated and acts as an antagonist of the Ang-1/Tie-2 axis, subsequently causing vascular destabilization and sensitizing blood vessels to synergistic effects of VEGF [39,40,41]. The current study demonstrated that eucalyptol promoted endothelial Ang-1/Tie-2 signaling and inhibited Ang-2 elevation in Aβ-exposed RVE cells and in the diabetic retina. Several reports have proposed the role of Ang proteins in DR [29,30,31], and several aspects of Ang signaling are being explored as novel therapeutic strategies. Thus, targeting the Ang-1/Tie-2 pathway or applying the Ang-2/VEGF combination may be a therapeutic option for eucalyptol to restore vascular stability and reduce abnormal neovascularization.

The Aβ deposition in AD weakens the vessel walls, which leads to cerebral hemorrhage [42]. In the same context, diabetes-associated Aβ accumulation in the retina destroys outer BRB, possibly resulting in eye defects such as macular edema [26]. Increased vascular permeability associated with angiogenesis, causes vision loss in DR and macular degeneration under diabetic conditions [43]. As earlier stated, eucalyptol inhibited the outer BRB breakdown responsible for the RPE detachment from the inner layer of Bruch’s membrane. The present study delineated the mechanistic outcomes of how eucalyptol affected inner BRB under diabetic conditions. While therapeutic targets for the outer BRB have been well studied due to its apparent relevance to retinitis pigmentosa and age-related macular degeneration, there is a lack of effective therapeutic drugs to maintain the high glucose/Aβ-induced RVE integrity. Eucalyptol inhibited the inner BRB damage through the preservation of the cell-to-cell adhesion proteins of VE-cadherin and occludin-1. In experiments with FITC-labeled BSA and FITC-dextran, eucalyptol blocked specific leakage from the inner BRB.

## 4. Materials and Methods

### 4.1. Materials

Fetal bovine serum (FBS), trypsin-EDTA, and penicillin–streptomycin were provided from Lonza (Walkersvillle, MD, USA). M199 media, mannitol, and D-glucose were obtained from Sigma-Aldrich Chemical (St. Louis, MO, USA), as were all other reagents, unless specifically stated elsewhere. Aβ protein was purchased from Calbiochem (San Diego, CA, USA). Rabbit polyclonal vascular endothelial cadherin (VE-cadherin) antibody was supplied by Abcam Biochemicals (Cambridge, UK). Goat polyclonal antibodies of Ang-1 and Ang-2, rabbit polyclonal antibodies of occludin-1 and phospho-PERK, and mouse monoclonal Aβ antibody were obtained from Santa Cruz Biotechnology (Santa Cruz, CA, USA). Goat polyclonal VEGF antibody was provided by R&D systems (Minneapolis, MN, USA). Rabbit polyclonal phospho-Tie-2 antibody was provided by Millipore Corporation (Temecula, CA, USA). Mouse monoclonal β-actin and rabbit polyclonal bcl-2 antibodies were provided by Sigma-Aldrich Chemical. Mouse bax antibody was obtained from BD Transduction Laboratories (West Grove, PA, USA). Antibodies of rabbit polyclonal caspase-12 and phospho-eIF2α, mouse monoclonal CHOP, and rabbit monoclonal ATF4 were obtained from Cell Signaling Technology (Danvers, MA, USA). Horseradish peroxidase-conjugated goat anti-rabbit immunoglobulin (Ig)G, goat anti-mouse, and donkey anti-goat IgG were purchased from Jackson Immumno Reserch Laboratories (West Grove, PA, USA). Essentially fatty acid-free bovine serum albumin (BSA) and skim milk were supplied by Becton Dickinson Company (Sparks, MD, USA).

### 4.2. Human RVE Cell Culture

Primary human RVE cells were obtained from Cell System Corporation (Kirkland, WA, USA). Cells were grown in M199 media containing 10% FBS, 100 U/mL penicillin, 100 μg/mL streptomycin, 2 mM glutamine, 0.75 μg/mL human epidermal growth factor, and 75 μg/mL hydrocortisone at 37 °C humidified atmosphere of 5% CO_2_ in the air. HRMVEC were sub-cultured at 90% confluence and used for further experiments within 10 passages. To induce a hyperglycemic condition, RVE cells were incubated in media containing 33 mM glucose, in comparison with normal media containing 5.5 mM glucose. In addition, HRMVEC were incubated for 3 days in media containing 5 μM Aβ in the absence and presence of 1–20 μM eucalyptol. Eucalyptol was dissolved in dimethyl sulfoxide for live culture with cells; a final culture concentration of dimethyl sulfoxide was <0.5%.

The human RVE cell viability was determined by assaying with MTT (3-(4,5-dimethylthiazol-2-yl)-2,5-diphenyltertrazolium bromide). Cells were incubated with 1 mg/mL MTT solution at 37 °C for 3 h, producing insoluble purple formazan that was dissolved in 250 μL isopropanol. Optical density was measured using a microplate reader at λ = 570 nm. Eucalyptol did not cause cytotoxicity at doses of ≤20 μM (Figure 1B). Thus, the current experiments employed eucalyptol in the range of 1–20 μM.

### 4.3. In Vivo Animal Experiments

Adult male db/db mice (C57BLKS/+Leprdb Iar; Jackson Laboratory, Sacramento, CA, USA) and their age-matched non-diabetic db/m littermates (C57BLKS/J; Jackson Laboratory, Sacramento, CA, USA) were used for the current study. Since db/db mice develop hyperglycemia at the age of 7–8 weeks, this study employed mice at 7 weeks of age. Mice were kept on a 12 h light/12 h dark cycle at 23 ± 1 °C with 50 ± 10% relative humidity under specific pathogen-free circumstances, fed a laboratory chow diet (CJ Feed, Seoul, Republic of Korea), and were supplied with water ad libitum at the animal facility of Hallym University. These animals were allowed to acclimatize for a week before commencing the experiments. Mice were divided into three subgroups (n = 9–10 for each subgroup). The first group of mice was non-diabetic db/m control mice, and db/db mice were divided into two groups. One group of db/db mice was daily supplemented with 10 mg/kg BW eucalyptol via gavage for 8 weeks. No mice were dead, and no apparent signs of exhaustion were observed in mice during the experimental intervention.

All experiments were approved by the Committee on Animal Experimentation of Hallym University and performed in compliance with the University’s Guidelines for the Care and Use of Laboratory Animals (hallymR1 2016-10).

The fasting blood levels of glucose and glycated hemoglobin HbA1C, a biomarker for diabetic complication, were measured every other week from mouse tail veins for 8 weeks. Eucalyptol decreased plasma levels of glucose and HbA1C markedly elevated in db/db mice [44]. The 24 h urine samples were measured in metabolic cages, in which urinary albumin secretion was reduced by supplementing eucalyptol to diabetic mice. In addition, plasma insulin level was highly reduced in eucalyptol-challenged mice [44].

### 4.4. Western Blot Analysis

Western blot analysis was conducted using whole-cell lysates prepared from RVE cells (3.5 × 10^5^ cells/dish) and eye tissue extracts. Whole-cell lysates and eye tissue extracts were prepared in a lysis buffer containing 1 M β-glycerophosphate, 1% β-mercaptoethanol, 0.5 M NaF, 0.1 M Na_3_VO_4_, and protease inhibitor cocktails. Human RVE cell lysates and eye tissue extracts containing equal amounts of proteins were electrophoresed on 8–15% SDS-PAGE and transferred onto a nitrocellulose membrane. Either 3% fatty acid-free BSA or 5% nonfat dry skim milk was added for 3 h to block nonspecific binding. The membrane was incubated overnight at 4 °C with each primary antibody of target proteins and washed in a Tris-buffered saline-Tween 20 for 10 min. The membrane was then incubated for 1 h with a secondary antibody of goat anti-rabbit IgG, goat anti-mouse IgG, or donkey anti-goat IgG conjugated to horseradish peroxidase. Each target protein level was determined by using immobilon Western chemiluminescent horseradish peroxidase substrate (Millipore Corporation, Burlington, MA, USA) and Agfa X-ray film (Agfa-Gevaert group, Mortsel, Belgium). Incubation with mouse monoclonal β-actin antibody was conducted as a comparative control.

### 4.5. Immunohistochemical Staining

For the immunohistochemical staining, eyes were extracted at the end of the experiments and fixed in 10% buffered formalin. The paraffin-embedded eye tissues were sectioned at 5 μm thickness, deparaffinized, and dehydrated with xylene and graded ethanol solutions. Eye tissue sections were incubated with a primary antibody of Aβ and bcl-2 overnight and FITC-conjugated anti-mouse IgG and FITC-conjugated anti-rabbit IgG, respectively. In addition, immunofluorescent histochemical staining for bax was done with Cy3-conjugated anti-mouse IgG. Nuclear staining was performed with 4′,6-diamidino-2-phenylindole (DAPI, Santa Cruz Biotechnology, Santa Cruz, CA, USA). Each slide was mounted in VectaMount mounting medium (Vector Laboratories, Burlingame, CA, USA). Images were taken using an optical Axiomager microscope system (Zeiss, Oberkochen, Germany). The protein levels of Aβ, bcl-2, and bax were quantified with an image analysis program from the microscope system.

### 4.6. Assay for DNA Fragmentation

For the detection of RVE cell apoptosis, terminal deoxynucleotidyl transferase dUTP nick end labeling (TUNEL) was conducted. Human RVE cells were seeded on a chamber slide and treated with 20 μM eucalyptol in Aβ-containing media. The DNA fragmentation was assayed by using a commercial DeadEnd™ Florometric TUNEL kit (Promega Corporation, Madison, WI, USA). The cells were fixed with 4% formaldehyde for 30 min and permeabilized with 0.2% Triton X-100. FITC-12-dUTP was added for 1 h at 37 °C. For the nuclear counterstaining, cells were treated with 4 mg/mL DAPI in a phosphate-buffered saline (PBS)-Tween 20 for 10 min. Representative fluorescent mages of each slide were taken and quantified using an Axiomager microscope system (Carl Zeiss, Oberkochen, Germany).

### 4.7. In Vitro Permeability Assay

The albumin permeability assay was performed by using transwell inserts with 8 μm pore size filters (Costar, Corning, NY, USA). Human RVE cells were plated on the inserts at (7 × 10^4^ cells/well) and treated with 20 μΜ eucalyptol in media containing 5.5 mM glucose, 33 mM glucose, or 5 μM Aβ for 3 days. Subsequently, 50 μg/mL FITC-labeled BSA was added to the upper inserts with serum-free media. After 2 h, the media in the lower compartment were collected, and the fluorescent intensity was measured by using Fluoroskan reader (Thermo Fisher Scientific, Waltham, MA, USA) at respective emission and excitation λ = 495 and λ = 520 nm.

### 4.8. FITC-Conjugated Dextran Retinal Flat Mounts

Mice were injected with 20 mg/mL FITC-dextran in PBS via the left ventricle. Immediately after the injection, the eyes were removed and fixed in a 4% paraformaldehyde solution for 10 min. The retinas were then flat-mounted and observed using an optical Axiomager microscope system.

### 4.9. Data Analysis

The data are presented as mean ± SEM for each treatment group in experiments. Statistical analyses were conducted using the Statistical Analysis Systems statistical software package version 6.12 (SAS Institute, Cary, NC, USA). Significance was determined by a one-way analysis of variance, followed by the Duncan range test for multiple comparisons. Differences were considered significant at *p* < 0.05.

## 5. Conclusions

The current study investigated the ability of eucalyptol to inner BRB collapse and subsequent vascular leakage in glucose/Aβ-loaded RVE cells and diabetic mouse eyes. Further, this study explored signaling pathways associated with vascular defects. Nontoxic eucalyptol deterred glucose-induced Aβ-mediated apoptotic loss of junction proteins of inner BRB with inhibition of caspase-12 activation and ER stress. Consistent with the culture results, oral administration of eucalyptol inhibited Aβ production and inner BRB disruption in diabetic eyes, then maintaining the barrier integrity. Further, Aβ-induced RVE cell apoptosis in the diabetic retina appeared to be attributed to the ER stress-induced PERK-eIF2α-ATF4 signaling pathway. Eucalyptol deterred increased angiogenesis via CHOP activation and VEGF elevation of RVE cells within the diabetic retina. The activation of Ang-1-Tie-2 signaling and dual inhibition of Ang-2/VEGF were promoted by treating eucalyptol to Aβ-exposed RVE cells and diabetic eyes. These findings support notions that eucalyptol improved vascular permeability, multiple scattered leakage, and aberrant angiogenesis in diabetic retinal vascular beds. Therefore, pharmacological intervention of eucalyptol could be a therapeutic option in counteracting diabetes-associated Aβ-mediated vascular abnormalities. Further studies on RVE barrier dysfunction are still needed regarding the dual activity of insulin and Aβ.

## Figures and Tables

**Figure 1 ijms-25-07826-f001:**
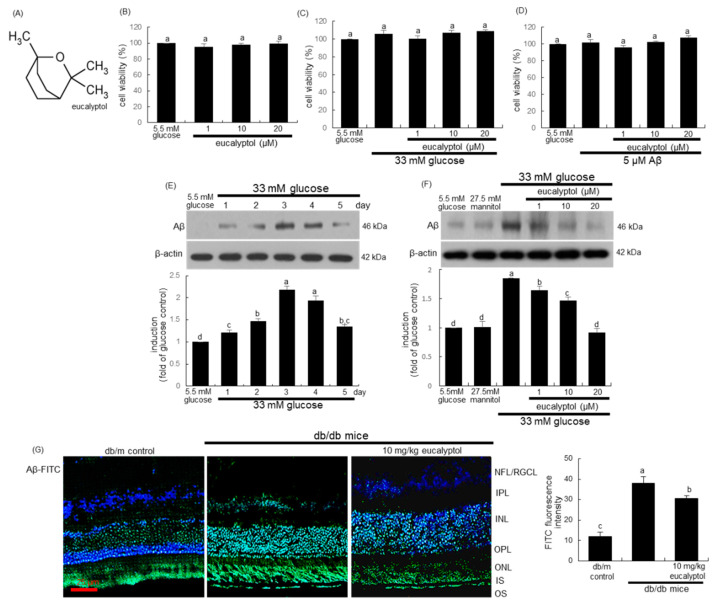
Chemical structure of eucalyptol (**A**), eucalyptol cytotoxicity (**B**), eucalyptol viability in high glucose-exposed human retinal microvascular endothelial (RVE) cells (**C**) and amyloid-β (Aβ)-exposed RVE cells (**D**), temporal responses of Aβ to glucose (**E**), and the inhibitory effect of eucalyptol on Aβ induction (**F**). Human RVE cells were cultured in 33 mM glucose media or with Aβ in the absence and presence of 1–20 μM eucalyptol for up to 5 days. Cells were also incubated with 5.5 mM glucose plus 27.5 mM mannitol as osmotic controls. Cell viability was measured by MTT assay (**B**–**D**). Bar graphs for viability (mean ± SEM, n = 5) were expressed as percent cell survival, compared to glucose control. Whole-cell lysates were subject to SDS-PAGE and Western blot with a specific antibody against Aβ (**E**,**F**). β-Actin protein was used as a cellular internal control of RVE cells. Bar graphs (mean ± SEM, n = 3) in the bottom panels represent densitometric results of upper blot bands. The db/db mice were orally supplemented with 10 mg/kg eucalyptol daily for 8 weeks. Immunohistochemical staining was performed to visualize the induction of Aβ in diabetic retinal tissues (**G**). Green fluorescein isothiocyanate (FITC)-conjugated secondary antibody was used for visualizing Aβ, being counterstained with 4′,6-diamidino-2-phenylindole for the blue nuclear staining. Each microphotograph (mean ± SEM, n = 3) was obtained by using an Axiomager microscope system. Scale bar = 25 μm. All the respective values do not share a common letter difference at *p* < 0.05.

**Figure 2 ijms-25-07826-f002:**
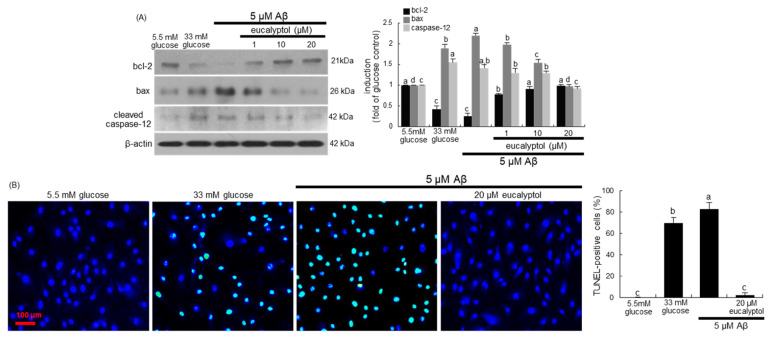
Effects of eucalyptol on induction of bcl-2, bax and cleaved caspase-12 (**A**), and apoptotic DNA fragmentation (**B**). Human retinal endothelial cells were treated with 1–20 μM eucalyptol for 3 days in culture media containing 5 μM amyloid-β (Aβ). Cells were also incubated in 33 mM glucose media. Whole-cell lysates were subject to SDS-PAGE and Western blot with a specific antibody against bcl-2, bax, or cleaved caspase-12 (**A**). β-Actin protein was used as a cellular internal control of RVE cells. The bar graphs (mean ± SEM, n = 3) represent quantitative results of blots in the left panel obtained from a densitometer. DNA fragmentation was measured with a TUNEL assay and nuclear counterstaining was done with blue-emitting fluorescent 4′,6-diamidino-2-phenylindole (**B**). Representative microphotographs were obtained by fluorescent microscopy with a fluorescein green filter. Fluorescence intensity was quantified by using an Axiomager microscope system. Scale bar = 100 μm. All the respective values not sharing a letter are different at *p* < 0.05.

**Figure 3 ijms-25-07826-f003:**
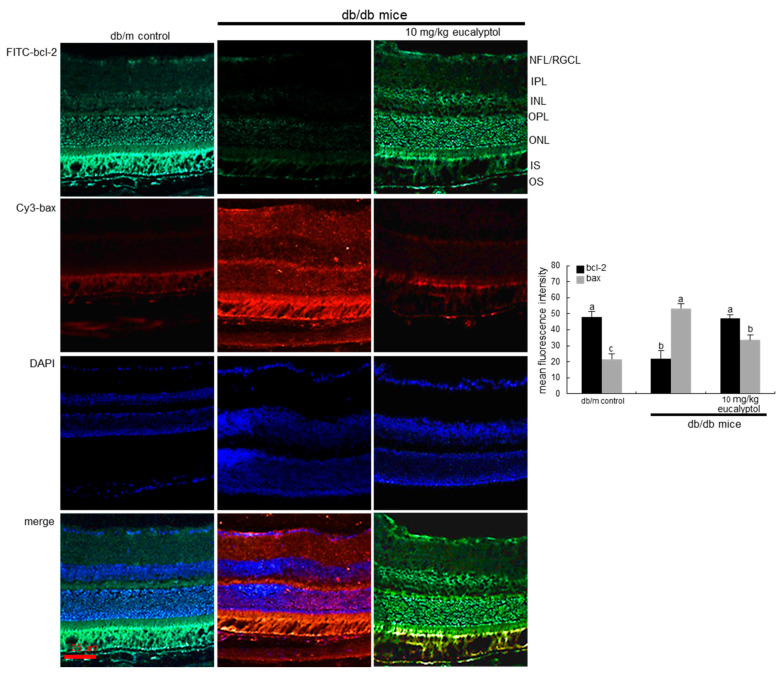
Reciprocal changes in retinal tissue levels of bcl-2 and bax in eucalyptol-supplemented db/db mice. The db/db mice were orally supplemented with 10 mg/kg eucalyptol daily for 8 weeks. Immunohistochemical staining of bcl-2 and bax was performed to visualize the induction of these proteins in diabetic retinas by using green fluorescein isothiocyanate (FITC)-conjugated secondary bcl-2 antibody and red Cy3-conjugated secondary bax antibody. Nuclear staining was conducted using blue 4′,6-diamidino-2-phenylindole (DAPI). Representative microphotographs were obtained using fluorescent microscopy with fluorescein green or red filters. Scale bar = 25 μm. The relative staining intensity was measured, and the respective values are expressed as mean ± SEM (n = 3 in each group). Values in bar graphs not sharing a letter indicate significant different at *p* < 0.05.

**Figure 4 ijms-25-07826-f004:**
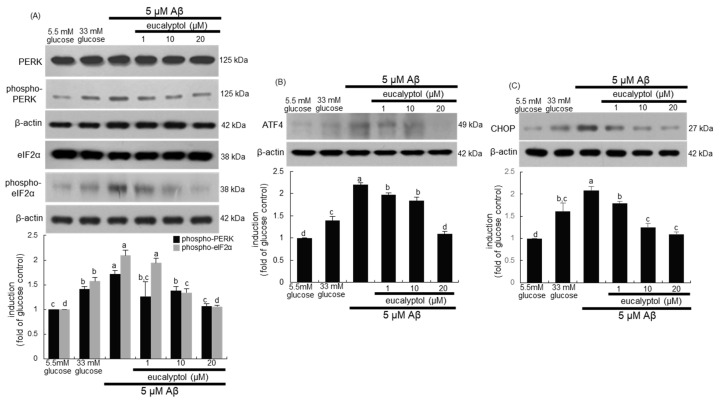
Western blot data showing inhibition of protein kinase R-like ER kinase (PERK) and eukaryotic initiation factor 2 alpha (eIF2α) activation (**A**), activating transcription factor-4 (ATF4) induction (**B**), and C/EBPα-homologous protein (CHOP) expression (**C**) by eucalyptol. Human retinal endothelial (RVE) cells were treated with 1–20 μM eucalyptol for 3 days in culture media containing 5 μM amyloid-β (Aβ). Cells were also incubated in 33 mM glucose media. Whole-cell lysates were subject to SDS-PAGE and Western blot with a specific antibody against PERK, phospho-PERK, elF2α, phospho-eIF2α, ATF4, or CHOP (**A**–**C**). β-Actin protein was used as a cellular internal control of RVE cells. The bar graphs (mean ± SEM, n = 3) represent quantitative results of blots in the upper panels obtained from a densitometer. Respective values not sharing a letter are different at *p* < 0.05.

**Figure 5 ijms-25-07826-f005:**
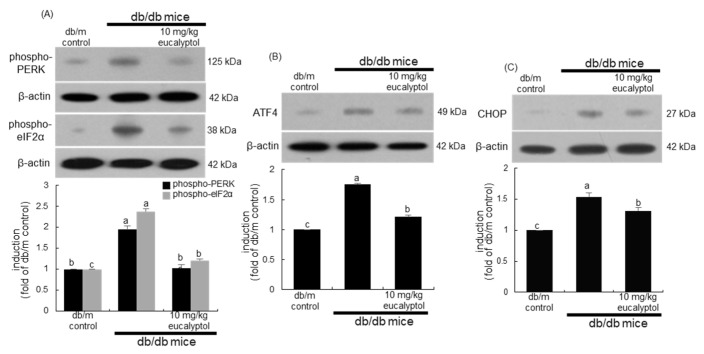
Inhibition of induction of phospho-protein kinase R-like ER kinase (PERK), phospho-eukaryotic initiation factor 2 alpha (eIF2α), activating transcription factor-4 (ATF4) and C/EBPα-homologous protein (CHOP) in eyes by eucalyptol. The db/db mice were orally supplemented with 10 mg/kg eucalyptol daily for 8 weeks. Whole eye tissue extracts were subject to SDS-PAGE and Western blot with a specific antibody against phospho-PERK, phospho-eIF2α, ATF4, or CHOP (**A**–**C**). β-Actin protein was used as an internal control of eye tissue extracts. The bar graphs (mean ± SEM, n = 3) represent quantitative results of blots in the upper panels obtained from a densitometer. Respective values not sharing a letter are different at *p* < 0.05.

**Figure 6 ijms-25-07826-f006:**
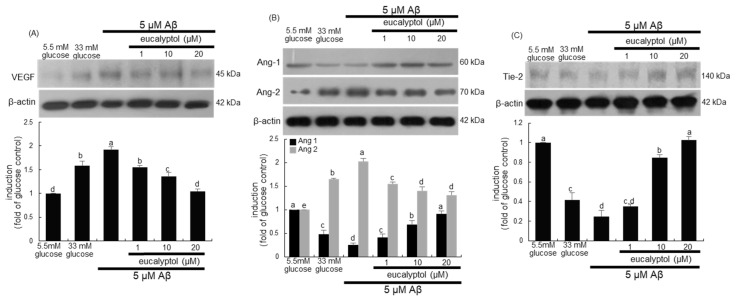
Effects of eucalyptol on expression of VEGF and induction of angiopoietin (Ang)-1, Ang-2, and Tie-2. Human retinal endothelial (RVE) cells were treated with 1–20 μM eucalyptol for 3 days in culture media containing 5 μM amyloid-β (Aβ). Cells were also incubated in 33 mM glucose media. Whole-cell lysates were subject to SDS-PAGE and Western blot with a specific antibody against VEGF, Ang-1, Ang-2, or Tie-2 (**A**–**C**). β-Actin protein was used as a cellular internal control of RVE cells. The bar graphs (mean ± SEM, n = 3) represent quantitative results of blots in the upper panels obtained from a densitometer. Respective values not sharing a letter are different at *p* < 0.05.

**Figure 7 ijms-25-07826-f007:**
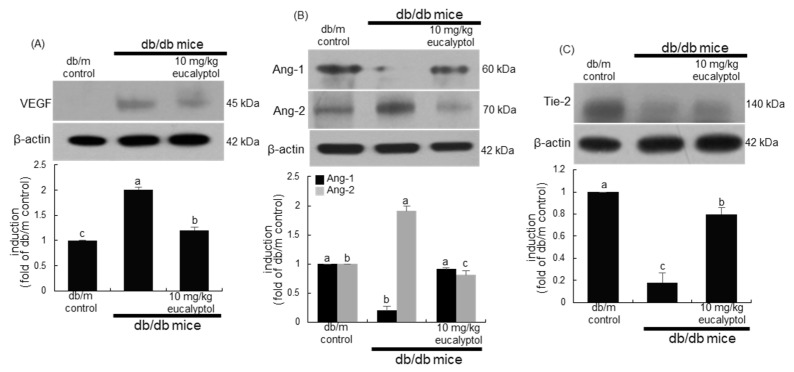
Western blot data showing eye tissue levels of VEGF, angiopoietin (Ang)-1, Ang-2, and Tie-2 in eucalyptol-supplemented db/db mice. The db/db mice were orally supplemented with 10 mg/kg eucalyptol daily for 8 weeks. Whole eye tissue extracts were subject to SDS-PAGE and Western blot with a specific antibody against VEGF, Ang-1, Ang-2, and Tie-2 (**A**–**C**). β-Actin protein was used as an internal control of eye tissue extracts. The bar graphs (mean ± SEM, n = 3) represent quantitative results of blots in the upper panels obtained from a densitometer. Respective values not sharing a letter are different at *p* < 0.05.

**Figure 8 ijms-25-07826-f008:**
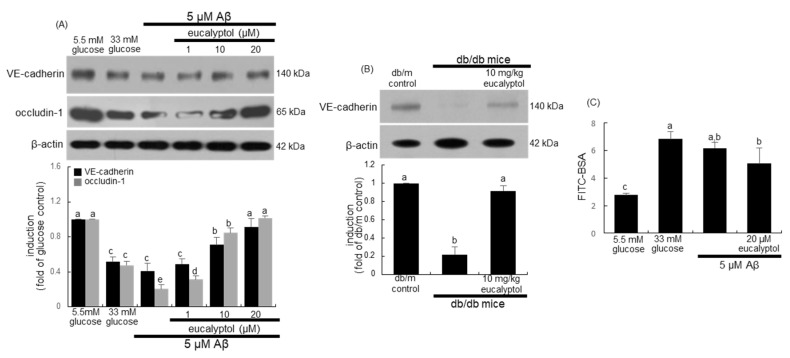
Effects of eucalyptol on induction of VE-cadherin and occludin-1 (**A**,**B**), and vascular permeability (**C**). Human retinal endothelial (RVE) cells were treated with 1–20 μM eucalyptol for 3 days in culture media containing 5 μM amyloid-β (Aβ). The db/db mice were orally supplemented with 10 mg/kg eucalyptol daily for 8 weeks. Whole-cell lysates and eye tissue extracts were subject to SDS-PAGE and Western blot with a specific antibody against VE-cadherin and occludin-1 (**A**,**B**). β-Actin protein was used as an internal control of RVE cells or eye tissue extracts. The bar graphs (mean ± SEM, n = 3) represent quantitative results of blots in the upper panels obtained from a densitometer. For the measurement of RVE injury, the albumin permeability assay was performed with fluorescein isothiocyanate (FITC)-labeled BSA (**C**). Respective values not sharing a letter are different at *p* < 0.05.

**Figure 9 ijms-25-07826-f009:**
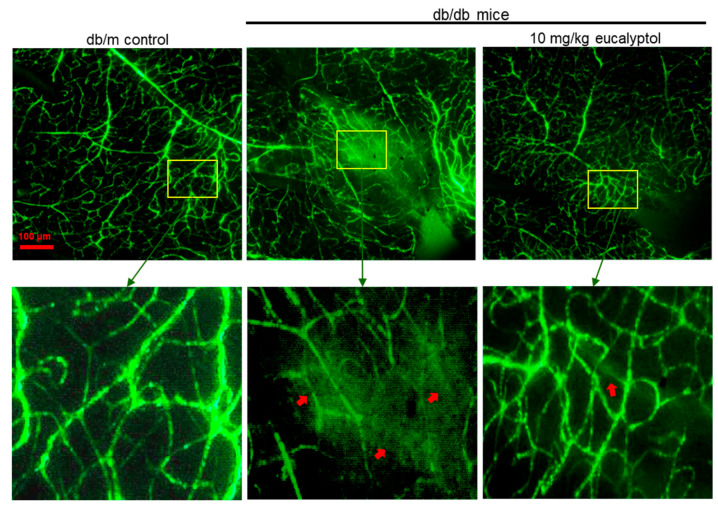
Inhibition of microvascular endothelial leakage by eucalyptol. The db/db mice were orally supplemented with 10 mg/kg eucalyptol daily for 8 weeks. The images below are magnified images within a square of each of the images above. Retinas were dissected, flat-mounted, and observed by confocal microscopy. Leakage of the green-colored fluorescent fluorescein isothiocyanate (FITC)-conjugated dextran appeared in diabetic retinal vessels (red arrows). Scale bar = 100 μm.

## Data Availability

Data is contained within the article.

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
