# Peer review of "Eucalyptol Ameliorates Retinal Microvascular Defects through Modulating ER Stress and Angiopoietin–Tie Signaling in Diabetic Eyes"

_ijms, 2024, doi:10.3390/ijms25147826_

Round 1

Reviewer 1 Report

Comments and Suggestions for Authors

In this study, the Eucalyptol’s therapeutic potential on DR has been investigated In vitro with RVE cells and in vivo with db/db mice. They demonstrated that Eucalyptol improves vascular permeability, junction protein expression, ER stress, cell apoptosis and amyloid-β formation in mouse retinas. They also carried out the in vitro studies using cell culture models, the results are consistent  to the in vivo results.

The manuscript is well written. The study is well designed. The conclusions are supported by the results. It would add new finding to the field of DR.

Minor:

1. regarding Human RVE cell culture, were the glucose, amyloid-β and Eucalyptol added at the same time and treated for the same duration? 

2. regarding RVE culture and TUNEL staining, were the cells maintained in the same culture condition (same amount of FBS, growth factor, and hydrocortisone etc.) from seeding to fixation?

Reviewer 2 Report

Comments and Suggestions for Authors

Overview: The authors have conducted lab-based research to understand if the Eucalyptol has any beneficial effect on the inner blood retinal barrier and microvasculopathy. The study was done on mice as well as glucose-exposed human retinal microvascular endothelial cells. They observed that the eucalyptol blocked endothelial cell damage by ameliorating ER stress by inhibiting Ang-2/VEGF mechanism. Ultimately the abnormal angiogenesis was blocked. Therefore, they recommend that eucalyptol as a new treatment strategy for diabetic microangiopathy.

This is a good study with useful information. I have a few points to clear. Please see the specific comments below and address them.

Specific comments:

Abstract

L14: Most readers may not know what the ER is! So, you may need to provide full form in the abstract.

L18: Is apoptosis the correct term here?

Introduction

L41: It should be exudative or serous retinal detachment. Later, it leads to tractional RD which has different mechanism. So mention exudative RD.

L49-61: You should focus on DR. Bringing this AD in suddenly does not seem relevant, disrupts the flow. You may move it to Discussion if you want to retain it.

Results

Figure 9: Please label each of the figures and elaborate the legend explaining all six images.

Discussion

Q: Any observation that you wish to obtain but could not, or any limitations you encountered in this study? If so, mention them.

Conclusion

The first sentence is incomplete. Please address it.

Methods

Q: What measures were taken to ensure that the loss of tight junction among endothelial cells in the Human RVE cell Culture was glucose-induced and not due to tissue being isolated?  

Q: How did your in vitro and in Vivo results compare?

Comments on the Quality of English Language

The quality of English language is high, except a few typo mistakes (see specific comments). 
